# Are the shareholding and trading behaviors of diverse investors affected by the relaxation of day trading?

**Wan-Hsiu Cheng[1], Yensen Ni[2]\*, Ting-Hsun Ho[2], Chia-Jung Chiang[2], Paoyu Huang[3], Yirung Cheng[2]**

**1** Department of Banking and Finance, Tamkang University, New Taipei, Taiwan, **2** Department of Management Sciences, Tamkang University, New Taipei, Taiwan, **3** Department of International Business, Soochow University, Taipei, Taiwan

\* ysniysni@gmail.com

**Data Availability Statement:** All relevant data are within the manuscript and its Supporting Information files.

## Abstract

The day trading in Taiwanese stock market expands considerably at the beginning of 2016, which increases the transactions of stocks consequently and sparks our interest in exploring the issue of day trading. In this study, we use the data of Taiwan Stock Exchange listed firms to investigate whether the day trading volume over total trading volume (hereinafter referred to as the day trading ratio) and the turnover ratio enhanced by the increase of day trading volume would affect the shareholding and trading behaviors of diverse institutional and individual investors. Unquestionably, we bring out several impressive findings. First, foreign institutional investors would not prefer holding or trading the stocks with high day trading ratios, whereas individual investors would prefer holding these kinds of stocks. We infer that this finding might result from the fundamental and the speculative concerns of these various investors. Second, domestic institutional investors and security dealers would prefer trading the stocks with high turnover ratios, but foreign institutional investors still lack of interest in trading these stocks, implying that the investment strategies would be dissimilar among various institutional investors. Since foreign institutional investors are regarded as the successful institutional investors in Taiwan, we argue that our revealed results may help market participants trace the behaviors of diverse investors, especially the foreign institutional investors, after day trading relaxation in Taiwan.

## Introduction

From the perspective of psychology, there is an environment-behavior theory that discusses the relationship between environmental change and behavior reaction. In theory, the environment is defined as the self-report measures used to assess subjective reactions to that environment [1] and then people evaluate their behavior with regard to the environment accordingly [2]. Besides, the intention is the best predictor of environmental behavior, whereas personal norms, attitudes, and perceived behavioral control have significant influences on behavioral intention [3]. In sum, the reaction is an attempt to find the results that make changes to the

**Funding:** The author(s) received no specific funding for this work.

**Competing interests:** The authors have declared that no competing interests exist.

individual as a new thing and to increase knowledge [4]. However, it seems to exist an interesting line of research links psychological influences and financial markets [5], which evokes our curiosity whether this psychological theory is still functioning for the investment activity. If it does work, we then might establish some investment models for investors to take suitable strategies in advance.

Inspired by the evidence that game rule change might cause behavioral change [6,7], we consider that it would be worthwhile to investigate the issue of environment-behavior change in the finance area. We speculate that the regulation change probably results in the change of investor's behavior in return. On Feb. 1, 2016, Taiwanese government unwinds the regulation of day trading in the stock market, which makes market participants believe that market liquidity may be improved and stock trading may have more flexibility. Even further, the relaxation may reduce the risk of overnight shareholdings for investors and may protect the interests of investors. Nevertheless, we argue that there are some issues still unanswered after the relaxation. For example, whether the individual and institutional investors, including foreign institutional investors, domestic institutional investors, and security dealers, would prefer trading or holding the stocks with higher day trading ratios (i.e., day trading ratio is defined as the day trading shares in a year over the total trading shares in a year), even higher turnover ratios (i.e., turnover ratio is defined as the total trading shares in a year over the total shares outstanding)? Does any difference exist for the preference of trading or holding these stocks among individual and institutional investors? These unresolved issues arouse our interest to organize this research.

We conduct this study to examine whether the relaxation of day trading regulation would change the diverse investors' behaviors, which is stemmed from the following motivations. First, we attempt to identify if the speculative individual investors might prefer investing stocks with higher day trading ratios and, in contrast, if the other individual deemed as the uninformative traders [8,9] might not prefer trading these stocks due to the difficulty of making profits [10]. Second, as realized with making more effort on fundamentals [11–13] instead of speculation [14–16], institutional investors should tend to decrease the shareholdings and trading volumes of stocks with higher day trading ratios. However, some institutional investors would prefer trading and holding the stocks with higher day trading ratios, especially for those having unreleased information [17–19]. Therefore, we endeavor to solve the puzzle of whether these institutional investors have this preference due to the liquidity concerns [20–22].

In this study, we reveal several impressive findings. First, foreign institutional investors might not prefer holding and trading the stocks with high day trading ratios nor high turnover ratios, whereas individual investors seem to have the opposite preference. We infer that this finding might result from the fundamental and the speculative concerns between foreign institutional investors and individual investors [23–25]. Second, we reveal that the enhancement of day trading after the unwinding might crowd out the day trading in credit on Taiwan stock markets. We deduce that this result might be caused by the substitution effect because the day trading volume through spot trading might decline the day trading volume via credit trading due to day trading permitted by credit trading (i.e., margin buying and short-selling the same stock in a day) only before the relaxation of day trading regulation. Third, domestic institutional investors and security dealers seem to prefer employing spread trading strategies and, as a result, might prefer trading the stocks with high turnover ratios due to the liquidity concern [26,27]. However, foreign institutional investors might not be interested in trading these stocks since buy-and-hold strategies are likely adopted by foreign institutional investors. Since foreign institutional investors are regarded as the successful investors in Taiwan [28,29], our revealed results may provide valuable information for market participants to learn, even to trace, the behaviors of diverse investors after the relaxation of day trading.

This study may contribute to the existing literature in several aspects. First, market participants might be interested in whether the behaviors (i.e. shareholding and trading behavior) of various investors would be affected by the relaxation of day trading regulation in Taiwan. Following a survey of relevant studies, we argue that examining as well as further explaining the shareholding and trading behaviors of diverse investors seem rarely concerned in the existing literature. Second, our revealed results might provide evidence for investors to obtain familiarity with the preference of trading or holding high day-trading-ratio stocks for diverse investors. With the familiarity of preference, market participants may realize the philosophies of different investors and then draw up their investment strategies. Third, we argue that the findings of this study might be beneficial for market participants in screening their holding stocks due to the fact that some investors (e.g. foreign institutional investors) somewhat outperform other investors (e.g. individual investors) [30–32].

The remainder of this study is organized as follows. Section 2 describes the literature review and hypotheses proposed. Section 3 introduces the methodology applied in this study. Section 4 presents the empirical results and analysis. Section 5 shows the discussion. The conclusion is revealed in Section 6.

## Literature review and hypotheses proposed

In order to familiarize ourselves with relevant studies, we conduct a survey of relevant literature in terms of intraday trading and day trading, institutional investors and individual investors, as well as diverse investors and day trading. Additionally, since investors' behaviors, such as trading and shareholding behaviors, are likely affected by the corporate governance and financial issues [29,33–39], we survey the literature in terms of investors' behaviors, corporate governance, and financial performance.

### Intraday trading and day trading

Generally, the intraday traders buy and sell financial instruments in the short term, typically within the same trading day [40]. In other words, intraday trading has an open position that is not left overnight [41] and its observations seem suitable for examining the volume–volatility relation [42]. The intraday data allows us to measure the immediate impact of a certain issue on stock returns and provides a finer picture that cannot be readily extracted from aggregating daily observations. In fact, the advantage of intraday information is particularly valuable if a certain issue influences the stock returns more significantly at certain trading hours, for example, at the market opening periods [5].

Day trading, defined as the purchase and sale of the same stock by the same investor on the same day [15], has garnered growing attention in the investment world, especially from investors who are attracted to the prospect of earning high profits by investing only a few hours per day [43]. Although day trading has gained extensive popularity among investors, its potential profit remains a controversial issue [44]. In fact, greater day trading activity leads to greater return volatility and the impact of a day trading shock dissipates gradually within an hour [45]. Besides, day trading increases the bid-ask spread, price depth, and stock volatility, indicating that day trading activities not only cause higher transaction costs and trading risk but also raise the market's ability to absorb price impact. In sum, the impacts of stock day trading on market quality are not all positive [46].

With regard to the literature, we find that previous studies mainly focus on the phenomena, characteristics, and information released of intraday trading. For example, the price volatility of intraday trading is higher at the open but lower at midday [47] and the spreads are higher at the beginning as well as the end of the day relative to the interior period [48]. Furthermore, the

price discovery occurs in intraday trading of E-Mini S&P 500 futures instead of the S&P 500 indices [49] and the initial price reaction to the declaration of dividend news on intraday stock is evident within a few minutes at most [50]. Besides, there are significant differences in average trading volume across trading hours in a day on New York Stock Exchange [51].

In sum, we find that most of the relevant studies are related to intraday volatility, the market microstructure of intraday trading, the information released of day trading, and intraday trading volume. However, whether investor behaviors (i.e. the behaviors of shareholding and trading) would be affected by the day trading volume enhancement after the relaxation of day trading regulation seems rarely explored in the relevant studies. Thus, we propose hypothesis 1 as shown below.

H1. The shareholding and trading behaviors of investors would be affected by the enhanced day trading volume after the relaxation of day trading regulation.

## Institutional and individual investors

Due to having perverse abilities of selecting securities and limited information, individual investors often sell the winning investments, hold the losing investments, make undiversified stock portfolios, and engage in repeating the past behaviors [52]. In fact, most of the trading losses for individual investors could be traced to their aggressive orders which likely cause the individual investors to suffer losses in stock markets [30]. On average, individual investors lose money from trading [15], which is buying stocks that earn subpar returns and selling stocks that earn strong returns [53]. However, some individual investors might adjust their behaviors, which improve their investment performance consequently [54].

Essentially, institutional investors, consisted of foreign institutional investors (FIIs), domestic institutional investors (DIIs), and security dealers (SDs), act as the vital roles in the business world [55]. For instance, firms with higher institutional ownerships are more likely to terminate poorly performing Chief Executive Officers and exhibit improvements in valuation over time [56]. Meanwhile, the enhanced ownership of FIIs would foster long-term investment in tangible, intangible, and human capital, which leads to significant increase in innovation output [57]. Moreover, the increased stability of institutional holding is related to the better firm performance [58] and a higher foreign-institutional-ownership firm would have better stock price performance [59]. Besides, a pronounced foreign ownership effect has been revealed, which is the stocks with high foreign ownership outperforming the stocks with low foreign ownership [28]. Whereas, the trades of institutional investors in stock markets seem to be information-driven since some institutional investors might be able to derive unreleased information in advance [60].

Additionally, we argue that the shareholding and trading behaviors of institutional investors might be affected by many issues of their investing firms, including liquidity, asymmetric information, corporate governance, and financial performance. For example, foreign institutional investors, who construct bridges between firms and decrease transaction costs as well as information asymmetry between bidder and target, act as the facilitators for corporate control in the international markets [61]. Besides, foreign institutions tend to rely on some corporate governance indicators when making investment decisions [62]. Moreover, the domestic funds and Qualified Foreign Institutional Investors tend to hold big firms with better accounting performances, higher B/M ratios, and higher price to cash flow ratios [63]. Indeed, institutional investors have strong preferences for the stocks of large firms with good governance [64].

The previous studies indicate that institutional investors would have better investment performance, which might result from the information asymmetry [65], the corporate governance [62], and the investing strategies employed [60,66]. However, we argue that the shareholding

and trading behaviors of individual as well as institutional investors impacted by the relaxation of day trading seem rarely concerned in the existing studies. Hence, hypothesis 2 is extracted as presented below.

H2. The shareholding and trading behaviors of individual and institutional investors would be impacted by the relaxation of day trading regulation.

## Diverse investors and day trading

Although individual day traders provide market liquidity by reducing the bid-ask spread, the temporary price volatility, and the temporal price impacts, most of them tend to behave as the irrational contrarian traders [67]. As a result, the trading of individual investors results in the large losses economically [30]. Aside from their aggressive orders as mentioned in the previous subsection, a large proportion of the individual investors are undiversified, and the extent of under-diversification is greater for investors who hold only retirement accounts [68]. Particularly, domestic individuals, accounting for the largest portion of total day trading activity, face substantial losses from day trading and individual day traders who trade more frequently and heavily are more likely to suffer such losses. On the contrary, domestic money managers and foreign institutional investors generally make substantial profits through day trading [69].

As for the institutional investors, it is evidenced that the trading volume increased by foreign institutions would boost subsequent returns [70]. In fact, a disproportionately large cumulative price impact of medium-size trades is mainly initiated by institutions [71], indicating that institutional investors might be deemed as the informed traders. Nevertheless, it is reported that only a few day traders can reliably earn positive abnormal returns [15].

We reveal that day trading issues might not closely link with the behavior of diverse investors after surveying the relevant studies. Therefore, we explore whether the shareholding and trading behaviors of institutional investors would be affected by day trading enhancement after the relaxation of day trading regulation at the beginning of 2016 in Taiwan. Besides, we explore whether the margin level (the proxy for individual shareholding) and credit trading (the proxy for individual trading) would be affected by this relaxation for the concern of comparison since credit trading (i.e. margin buy and short-selling individual stocks) is only permitted for individual investors in Taiwan, so credit trading and margin level are employed as the proxies of individual trading and individual shareholding in this study. Thus, credit trading and margin level are deemed as the behaviors of individual investors in Taiwan.

In consequence, the hypothesis 3 and 4 are brought out as described below.

H3. The shareholding and trading behaviors of institutional investors would be affected by the relaxation of day trading regulation.

H4. The margin level and credit trading would be affected by the relaxation of day trading regulation.

## Investors' behaviors, corporate governance, and financial statements

In essence, corporate governance would be regarded as an important index for institutional investors. Firms with better corporate governance tend to be well-managed and have preeminent performance, which would increase the firms' values and might attract foreign investments [36]. In addition, firms with higher institutional ownership are more likely to terminate poorly performing

CEOs and exhibit improvements in valuation over time [56]. Interestingly, the fraction of institutional shareholding would improve the quality of governance structure for the firm [34].

Moreover, we argue that the ownership and board structure have critical impacts on institutional shareholding. For instance, the board size significantly relates to earnings management [72]. In general, a firm with a well-functioned board structure would have a positive impact on its value, which results in the increase of institutional shareholding [73]. Besides, firm value enhanced by monitoring board activities from outsiders would appeal to institutional investments [74]. Meanwhile, firms with greater insider ratios could reduce firm risk and perform better even in highly volatile environments [75].

In addition, financial statements play the significant roles for investors because much of information is contained in these statements. For instance, foreign institutions show their preference to invest in the firms with abundant cash holding as presented on the statement [37]. Furthermore, stock prices are increased when enterprises exhibit improved asset management [76], while corporate governance and financial performance are related to the shareholding change of foreign institutions [29]. Moreover, the financial risk of a firm may decline if the firm has a relatively lower debt ratio [77]. In sum, we claim that the investment plans of investors are deeply influenced by the financial performances.

Nevertheless, we find that the relationship between corporate governance and individual investors seems seldom examined in the relevant studies, which might result from individual investors paying less attention to the issue of corporate governance, as compared with institutional investors [78,79]. Thus, we endeavor to exploit more information about the relationships among the behaviors of individual investors, the corporate governance, and the financial statement in this study.

H5. The behaviors of individual investors would relate to the issues of corporate governance and financial statement.

## Methodology

### Data

Day trading stocks is permitted from 2014 in Taiwan stock exchange; however, there are only 377 stocks (only 25% stocks listed on TWSE) permitted over the period of 2014–2015. Due to the data consistency concerns, we select the data after day trading stocks permitting considerably (i.e. 1432 stocks covering over 95% stocks listed on TWSE) at the beginning of 2016. As a result, we select the data of the year 2016 and 2017. By summing up the daily day trading volume over one year for each stock, we derive the yearly day trading volume. Although day trading is regarded as a short-term activity in a day, summarizing the daily trading volume into the yearly trading volume might mix up or offset some of the important short-term information. However, the yearly data would be more appropriately related to firm characteristics and beneficial for exploring the factors which affect the shareholding and trading behaviors of diverse investors.

In this study, the ratio of day trading volume over total share outstanding could be equal to day trading ratio (i.e. day trading volume over total trading volume) multiplied by the turnover ratio (i.e. total trading volume over total share outstanding). The former could be measured as mentioned above, and the latter could be derived from Taiwan Economic Journal (TEJ). Thus, based on surveying the relevant literature for controlling variables in terms of board structure, financial statement, and others, we then explore whether the shareholdings or trading volume of diverse investors would be affected by either day trading ratio or turnover ratio after incorporating such controlling variables in this study since we argue that a stock with a higher day

trading ratio and turnover ratio might enhance the liquidity of the stock, thereby likely affecting the shareholding or trading volume of diverse investors.

Additionally, we classify the variables employed in this study into shareholding category, trading volume category, board structure category, and financial category. Because margin trading and short selling stocks are permitted only for individual investors in Taiwan, we deem margin level and credit trading ratio as the individual behaviors in Taiwan as well as in China [80–82]. Therefore, we employ margin level and credit trading ratio as the proxies of individual shareholding and trading volume in this study. As a result, the shareholding category includes foreign shareholding ratio, institutional shareholding ratio, dealer shareholding ratio, and margin level (i.e., the proxy for individual shareholding ratio). The trading volume category contains foreign trading ratio, institutional trading ratio, dealer trading ratio, and credit trading ratio (i.e., the proxy for individual trading ratio). The board structure category is composed of directors' holding ratio, top 10 shareholding ratio, managers' holding ratio, directors' pledge ratio, CEO duality, board size, and independent director dummy. The financial category consists of debt ratio, asset turnover ratio, net profit ratio, cash flow ratio, and others which are electronic dummy and firm scales. In order to be familiar with these variables used in this study, we define the variables as shown in Table 1.

## Descriptive statistics

By utilizing 1,577 observations of TWSE listed firms over the period of 2016–2017 as our samples, Table 2 presents the descriptive statistics, including the number of observations, means, medians, standard deviations, minima, and maxima of the variables employed in this study. Besides, we also expand our data to 2019, and the results are similar to those shown in Tables 3 and 4.

Table 2 shows that the maxima as well as minima of diverse shareholding and trading ratios are rather wide, especially for the shareholding ratio and trading ratio of FIIs. The numbers indicate that diverse investors might have different preferences in trading and holding the stocks listed on TWSE. In addition, we reveal that the range of either day trading ratio or turnover ratio is rather broad, indicating that investors might have big different preferences in trading or holding stocks.

Besides, Table 2 describes that the minimum director holding ratio is close to 0, the maximum directors' pledge ratio is up to 97.69%, and the maximum debt ratio reaches 96.54%, implying that certain firms listed on the TWSE might have corporate governance issues.

## Model

We set the models to explore whether either various shareholding ratios or diverse trading volume ratios would be affected by day trading ratio and turnover ratio in Models (1A)–(4A) which represent the shareholding behaviors or Models (1B)–(4B) that denote the various trading behaviors, respectively.

$$\begin{aligned} Y_{j,i,t} = \ &\beta_0 + \beta_1 \text{ Day trading ratio}_{i,t} + \beta_2 \text{ Turnover ratio}_{i,t} + \beta_3 \text{ Directors' shareholding ratio}_{i,t} \\ &+ \beta_4 \text{ Top 10 shareholders' holding ratio}_{i,t} + \beta_5 \text{ Managers' shareholding ratio}_{i,t} \\ &+ \beta_6 \text{ Directors' pledge ratio}_{i,t} + \beta_7 \text{ CEO duality}_{i,t} + \beta_8 \text{ Board size}_{i,t} \\ &+ \beta_9 \text{ Independent director dummy}_{i,t} + \beta_{10} \text{Debt ratio}_{i,t} + \beta_{11} \text{ Assets turnover ratio}_{i,t} \\ &+ \beta_{12} \text{ Net profit ratio}_{i,t} + \beta_{13} \text{ Cash flow ratio}_{i,t} + \beta_{14} \text{ Electronic dummy}_{i,t} \\ &+ \beta_{15} \text{ firm scales}_{i,t} + \varepsilon_{i,t} \text{ for } j = \ 1 \ to \ 8 \ \dots. \end{aligned} \qquad (1A)-(4B)$$

where $Y_{j,i,t}$ is foreign shareholding ratio for $j = 1$ as Model (1A), domestic institutional

**Table 1. Definitions of variables.**

| Variables | Definitions |
|---|---|
| **Shareholding behavior variables** | |
| Foreign shareholding ratio | Foreign institutional shareholding over total shares outstanding |
| Institutional shareholding ratio | Domestic institutional shareholding over total shares outstanding |
| Dealer shareholding ratio | Security Dealers' shareholding over total shares outstanding |
| Margin level | Margin shares over total shares outstanding * 0.25 |
| **Trading behavior variables** | |
| Foreign trading ratio | Foreign institutional trading volume in a year over total trading volume in a year |
| Institutional trading ratio | Domestics Institutional trading volume in a year over total trading volume in a year |
| Dealer trading ratio | Security dealers trading volume in a year over total trading volume in a year |
| Credit trading ratio | Credit trading volume in a year over total trading volume in a year |
| **Turnover variables** | |
| Day trading ratio | Day trading shares in a year over total trading shares in a year |
| Turnover ratio | Total trading shares in a year over shares outstanding |
| **Board structure variables** | |
| Directors' holding ratio | Total directors' shareholdings over total shares outstanding |
| Top 10 holding ratio | Top 10 shareholders' shareholdings over total shares outstanding |
| Managers' holding ratio | Total Managers' shareholdings over total shares outstanding |
| Directors' pledge ratio | Directors' pledged shares over total directors' shareholdings |
| CEO duality | Set to 1 if the chairman of a firm is the CEO; otherwise, set to 0 |
| Board size | Total number of directors on the board |
| Independent director dummy | Set to 1 if a firm recruits independent directors; otherwise, set to 0 |
| **Financial variables** | |
| Debt ratio | Total debts over total assets |
| Asset turnover ratio | Total sales over total assets |
| Net profit ratio | Net profit over total sales |
| Cash flow ratio | Operating cash flow over current liabilities |
| **Other relevant variables** | |
| Electronic dummy | Set to 1 if a firm falling into the electronic industry; otherwise, set to 0 |
| Firm scales | ln (market value) |

shareholding ratio for $j = 2$ as Model (2A), dealer holding ratio for $j = 3$ as Model (3A), and margin level for $j = 4$ as Model (4A), foreign trading ratio for $j = 5$ as Model (1B), institutional trading ratio for $j = 6$ as Model (2B), dealer trading ratio for $j = 7$ as Model (3B), or credit trading ratio for $j = 8$ as Model (4B). In sum, Model (1A) to (4A) represent the diverse shareholding behaviors; whereas, Model (1B) to (4B) denote the various trading behaviors.

Furthermore, in order to examine the existence of multicollinearity problems for these independent variables, the variance inflation factor (VIF) tests are used in the beginning. The VIF is an indicator of the severity of multicollinearity which is not considered as a severe problem if the VIF value is less than ten [80]. The results show that all of VIF values are less than three, indicating that multicollinearity issues are not serious in this study.

Furthermore, we concern whether the endogenous problem exists in our models since the shareholding ratio and trading ratio may affect day trading ratio and turnover ratio as well. Therefore, by using instrument variables estimated by both the two-stage least squares and generalized methods of moment approaches, we find that day trading ratio and turnover ratio could be regarded as the exogenous variables as revealed by the insignificant Hausman

**Table 2. Descriptive statistics.**

| Variables | Obs. | Mean | Median | SD | Max | Min |
|---|---|---|---|---|---|---|
| Foreign shareholding ratio (%) | 1577 | 13.96100 | 8.59000 | 15.66700 | 89.21000 | 0.01000 |
| Institutional shareholding ratio (%) | 1577 | 0.48000 | 0.00000 | 1.38900 | 20.21000 | 0.00000 |
| Dealer shareholding ratio (%) | 1577 | 0.11000 | 0.01000 | 0.58300 | 10.00000 | 0.00000 |
| Margin level ratio (%) | 1577 | 11.79900 | 7.67000 | 12.97400 | 98.81000 | 0.00000 |
| Foreign trading ratio (%) | 1577 | 14.04000 | 9.07000 | 15.13700 | 84.20000 | 0.00000 |
| Institutional trading ratio (%) | 1577 | 0.69400 | 0.00000 | 2.85800 | 32.14000 | 0.00000 |
| Dealer trading ratio (%) | 1577 | 1.42300 | 0.05000 | 2.98000 | 24.69000 | 0.00000 |
| Credit trading ratio (%) | 1577 | 1.16000 | 0.23000 | 2.69700 | 34.31000 | 0.00000 |
| Day trading ratio | 1577 | 15.62500 | 10.31000 | 15.70000 | 103.08800 | 0.04400 |
| Turnover ratio | 1577 | 122.59600 | 64.89800 | 168.79400 | 2404.45500 | 0.61340 |
| Debt ratio | 1577 | 41.78300 | 41.88000 | 17.67400 | 96.54000 | 0.90000 |
| Asset turnover ratio | 1577 | 0.79800 | 0.70000 | 0.56600 | 5.35000 | 0.01000 |
| Net profit ratio (%) | 1577 | 4.07500 | 5.49000 | 57.84100 | 465.83000 | -1613.22000 |
| Cash flow ratio (%) | 1577 | 28.40800 | 21.69000 | 72.32400 | 719.74000 | -1463.88000 |
| Directors' holding ratio (%) | 1577 | 21.86900 | 17.93000 | 15.40000 | 94.55000 | 0.01000 |
| Top 10 shareholding ratio (%) | 1577 | 1.51100 | 0.27000 | 4.55300 | 77.83000 | 0.00000 |
| Managers' holding ratio (%) | 1577 | 24.00400 | 21.72000 | 12.78700 | 73.25000 | 0.51000 |
| Directors' pledge ratio (%) | 1577 | 8.05400 | 0.00000 | 15.55400 | 97.69000 | 0.00000 |
| .CEO duality | 1577 | 0.29900 | 0.00000 | 0.45800 | 1.00000 | 0.00000 |
| Board size | 1577 | 7.96000 | 7.00000 | 2.26800 | 20.00000 | 2.00000 |
| Independent director dummy | 1577 | 0.93900 | 1.00000 | 0.23900 | 1.00000 | 0.00000 |
| Electronic dummy | 1577 | 0.46300 | 0.00000 | 0.49800 | 1.00000 | 0.00000 |
| Firm scales | 1577 | 15.79700 | 15.60200 | 1.32800 | 22.50600 | 12.80800 |

statistics. Meanwhile, we exclude outliers to prevent our results from being twisted by outliers, which might derive more objective results for the study.

## Empirical findings and analysis

### Results of shareholding behaviors for diverse investors

In this study, we explore whether the trading and the shareholding behaviors of diverse investors, including FIIs, DIIs, SDs, and individual investors, would be affected by day trading ratio and turnover ratio after incorporating board structure, financial statements, and other controlling variables. Table 3 reports the empirical results with the Petersen models.

Table 3 reveals that day trading ratio and turnover ratio negatively relate to the foreign shareholding ratio ($\beta$ = -0.01892**, $P< 0.05$; $\beta$ = -0.00357, $P<0.05$) but positively relate to the market level ratio ($\beta$ = 0.09990***, $P<0.01$; $\beta$ = -0.03887, $P<0.01$). These results indicate that FIIs might not prefer holding the stocks with higher day trading ratio and turnover ratio but individual investors prefer holding these stocks. Besides, we find that institutional investors, except FIIs, prefer holding the stocks with higher turnover ratio, indicating that diverse institutional investors might not have the same preference.

In addition, we also concern whether turnover ratio could play a moderating role by incorporating an interactive variable of "day trading ratio" multiplied by "turnover ratio" into regressions to test its moderating effect, which might enrich this study. However, due to the multicollinearity issue, we might not be able to incorporate this interaction term. We argue that the issue might result from the variables defined in this study (i.e. day trading ratio is

**Table 3. Results of shareholding behaviors for diverse investors.**

| Dependent variable / Independent variable | (1A) Foreign Institutional Shareholding ratio | (2A) Domestic Institutional shareholding ratio | (3A) Security Dealer shareholding ratio | (4A) Margin level ratio |
|---|---|---|---|---|
| Day trading ratio | -0.01892** | 0.00932 | 0.00084 | 0.09990*** |
| | (0.00846) | (0.00799) | (0.001171) | (0.03635) |
| Turnover ratio | -0.00357** | 0.00151** | 0.00030** | 0.03887*** |
| | (0.00163) | (0.00085) | (0.00014) | (0.00858) |
| Directors' shareholding ratio | 0.11589*** | 0.00196 | 0.00074 | -0.06968*** |
| | (0.02395) | (0.00170) | (0.00143) | (0.01289) |
| Top 10 shareholding ratio | 0.09584 | 0.00027 | -0.00160 | -0.10017** |
| | (0.07654) | (0.00320) | (0.00122) | (0.04621) |
| Mangers' shareholding ratio | 0.21044*** | 0.00441*** | -0.00001 | -0.04632** |
| | (0.04083) | (0.00157) | (0.00142) | (0.01916) |
| Directors' pledge ratio | -0.03252* | -0.00169 | 0.00280 | 0.04746 |
| | (0.01787) | (0.00183) | (0.00214) | (0.03549) |
| CEO duality | 2.03262 | -0.18377 | -0.03798 | 1.65187 |
| | (1.63312) | (0.14750) | (0.02956) | (0.99822) |
| Board size | -0.41376** | -0.04849*** | -0.00100 | -0.01564 |
| | (0.16797) | (0.00775) | (0.00432) | (0.08551) |
| Independent director dummy | 1.40542*** | 0.14284 | 0.04929*** | -1.06428 |
| | (0.18190) | (0.10190) | (0.01216) | (1.18703) |
| Debt ratio | -0.00470 | -0.00162 | -0.00178* | -0.00882 |
| | (0.01946) | (0.00144) | (0.00093) | (0.01286) |
| Asset turnover ratio | 3.97367*** | 0.27124*** | 0.02856 | -1.12283 |
| | (0.69410) | (0.05306) | (0.02076) | (1.08365) |
| Net profit ratio | 0.00210 | 0.01147*** | 0.00068 | -0.05357 |
| | (0.01954) | (0.00274) | (0.00049) | (0.04346) |
| Cash flow ratio | 0.02466*** | 0.00283*** | 0.00136*** | 0.00023 |
| | (0.00677) | (0.00061) | (0.00051) | (0.00413) |
| Electronic dummy | 3.49586*** | 0.02766 | -0.06877** | 0.53605 |
| | (0.69410) | (0.03381) | (0.02931) | (0.57776) |
| Firm scales | 6.95080*** | 0.18420*** | -0.00815 | -2.58014*** |
| | (0.30541) | (0.01780) | (0.00771) | (0.44860) |
| Constant | -101.8849*** | -2.88669*** | 0.16446 | 50.50962*** |
| | (4.385597) | (0.27228) | (0.11804) | (4.66163) |
| Outliers excluded | Yes | Yes | Yes | Yes |
| $R^2$ | 0.4448 | 0.1583 | 0.0324 | 0.4942 |
| Coefficient estimates | OLS | OLS | OLS | OLS |
| Standard errors | Cluster F & T | Cluster F & T | Cluster F & T | Cluster F & T |

We explore whether the shareholding ratio of FIIs, DIIs, and SDs as well as margin level ratio (the proxy of individual shareholding) would be affected by day trading ratio and turnover ratio after controlling the variables in terms of board structure, financial statements, and others. The results are shown in Equation (1A)–(4A). The t-statistics are based on the standard errors adjusted by the two-way clusters existed in firm and year [83] for Equation (1A)–(4A). Statistical significance values at the 10%, 5%, and 1% levels are denoted by *, **, and ***, respectively.

**Table 4. Results of trading behaviors for diverse investors.**

| Dependent variable / Independent variable | (1B) Foreign Institutional Trading ratio | (2B) Domestic Institutional Trading ratio | (3B) Dealer Trading ratio | (4B) Credit Trading ratio |
|---|---|---|---|---|
| Day trading ratio | -0.07144*** | 0.00493 | 0.00029 | -0.03064*** |
| | (0.00417) | (0.01035) | (0.01675) | (0.00131) |
| Turnover ratio | -0.00884*** | 0.00093 | 0.00251 | 0.003971 |
| | (0.00075) | (0.00114) | (0.0022) | (0.00318) |
| Directors' holding ratio | 0.07497*** | 0.01418*** | 0.00440 | 0.006468 |
| | (0.00982) | (0.00319) | (0.00401) | (0.00423) |
| Big Ten holding ratio | 0.05066 | 0.04201 | -0.01041* | -0.01055 |
| | (0.07116) | (0.07234) | (0.00630) | (0.00682) |
| Mangers' holding ratio | 0.02548** | 0.004123 | -0.00870 | -0.00243 |
| | (0.012631) | (0.00295) | (0.00559) | (0.01164) |
| Directors' pledge ratio | -0.05276*** | -0.00090 | 0.00024 | 0.004684 |
| | (0.01235) | (0.00214) | (0.00250) | (0.00361) |
| CEO duality | -0.08506 | -0.01148 | 0.04781 | -0.00486 |
| | (0.91225) | (0.09500) | (0.08464) | (0.02423) |
| Board size | -0.09267 | -0.05724*** | -0.03251 | -0.06080 |
| | (0.23119) | (0.02149) | (0.02647) | (0.04354) |
| Independent director dummy | 0.81073*** | 0.34686* | -0.07570 | -0.63264 |
| | (0.31113) | (0.20979) | (0.13718) | (0.62661) |
| Debt ratio | -0.00012 | -0.00409 | 0.00473 | -0.00423 |
| | (0.00776) | (0.01150) | (0.00368) | (0.00368) |
| Asset turnover ratio | 1.24495*** | 0.57200** | 0.58263*** | -0.31465* |
| | (0.38666) | (0.26998) | (0.13882) | (0.17628) |
| Net profit ratio | -0.04032 | 0.01156** | 0.01678*** | -0.02432*** |
| | (0.02675) | (0.00502) | (0.00359) | (0.00771) |
| Cash flow ratio | -0.00018 | 0.003538 | -0.001599 | -0.00181 |
| | (0.01215) | (0.003652) | (0.00365) | (0.00214) |
| Electronic dummy | 2.39214** | -0.25324*** | -0.120515 | 0.55004*** |
| | (1.17047) | (0.06516) | (0.20699) | (0.07801) |
| Firm scales | 8.00238*** | 0.31198*** | 0.77658*** | -0.05174* |
| | (1.22005) | (0.04153) | (0.08686) | (0.03621) |
| Constant | -109.3987*** | -5.07697*** | -11.34104*** | 2.71594*** |
| | (16.59454) | (0.66994) | (1.50856) | (0.66854) |
| Outliers excluded | Yes | Yes | Yes | Yes |
| $R^2$ | 0.5041 | 0.0586 | 0.1705 | 0.0596 |
| Coefficient estimates | OLS | OLS | OLS | OLS |
| Standard errors | Cluster F & T | Cluster F & T | Cluster F & T | Cluster F & T |

We explore whether the trading ratio of FIIs, DIIs, and SDs as well as credit trading ratio (the proxy of individual trading) would be affected by day trading ratio and turnover ratio after controlling the variables in terms of board structure, financial statements, and others. The results are shown in Equation (1B)–(4B). The t-statistics are based on the standard errors adjusted by the two-way clusters existed in firm and year [83] for Equation (1B)–(4B). Statistical significance values at the 10%, 5%, and 1% levels are denoted by *, **, and ***, respectively.

defined as day trading shares in a year over total trading shares in a year, turnover ratio is defined as total trading shares in a year over shares outstanding, and interaction terms would be day trading shares in a year over shares outstanding in a year).

Furthermore, we disclose that institutional investors would prefer holding the stock of a firm with a well-functioned board structure, such as higher directors' shareholding, higher managers' shareholding, as well as small board size, and better financial performance, such as higher asset turnover ratio and higher cash flow ratio. Moreover, we illustrate that FIIs prefer holding large-scale electronic firms because of the evidence that FIIs hold about 80% shareholding of Taiwan Semiconductor Manufacturing Company, the most heavy-weighted stock in Taiwan [28,29].

However, we reveal that individual investors (i.e. the margin level ratio employed as the proxy of individual shareholding) prefer holding the stock of a small firm without a well-functioned board structure (e.g. lower directors' shareholding ratio and lower managers' ratio). We infer that numerous individual investors might prefer holding speculative stocks without solid fundamentals since the stock price of a small firm without a well-functioned board structure is likely manipulated by market participants [84]. In addition, because the permission for margin trading in Taiwan is only for individual investors, we deduce that individual investors might not outperform the institutional investors, which probably results from the shareholding behaviors of individual investors differing from those of institutional investors [85,86].

## Results of trading behaviors for diverse investors

Aside from the exploration of shareholding behaviors of diverse investors, we also examine whether trading behaviors of diverse investors (i.e. the trading ratio of FIIs, DIIs, SDs, and individual investors) would be impacted by the day trading ratio as well as the turnover ratio and further compare the similarities as well as differences of these two behaviors (i.e. shareholding and trading behaviors) among diverse investors.

Table 4 reveals that the day trading ratio and the turnover ratio are all negatively related to foreign trading ratio. These results indicate that FIIs might not prefer trading the stocks with higher turnover ratios, even higher day trading ratios. On the contrary, FIIs prefer trading weighted stocks, large-scales stocks, whose turnover ratios and day trading ratios are rather lower because of the considerable outstanding shares.

However, we reveal that the day trading ratio has a negative relation with the credit trading ratio (i.e. the proxy of individual trading behaviors), which is somewhat different from that the day trading ratio positively relates to the margin level ratio (i.e. the proxy of individual shareholding behaviors) as shown in Tale 3. This finding might result from that the day trading volume by margin trading is probably substituted by spot trading because, before the relaxation of day trading regulation, day trading stocks is permitted by adopting margin trading only. In other words, the substitution effect between margin day trading and spot day trading might account for this outcome.

Regarding to the effects of board structure, financial performance, and other controlling variables on trading behaviors of investors, we find that institutional investors would prefer trading the stocks with a well-functioned board structure and better financial performance, such as the stocks of the firm with higher directors' shareholding ratio, small board size, higher asset turnover ratio, and higher firm scales. Besides, FIIs incline to trade the stocks of the firm with lower directors' pledge and large scale, indicating that FIIs might concern the issues of liquidity and corporate governance for trading stocks listed on TWSE. In addition, because higher stock price volatilities are likely to exist in the stocks of small-size firms [87], such as small electronic stocks in Taiwan, the stocks with high credit trading ratios might not have solid fundamentals (e.g. lower net profit ratio and lower asset turnover ratio) or speculative characteristics (e.g. small electronic stocks), which probably results in not easy to profit from credit trading stocks for individual investors.

While comparing the similarities and differences of shareholding and trading behaviors among diverse investors, we discover that institutional investors, except FIIs, would prefer trading the stocks with higher turnover ratio. This finding indicates that the trading behaviors might not be the same among diverse institutional investors. Conversely, aside from the above-mentioned difference, we reveal that most of the diverse institutional investors prefer either holding or trading the stocks of the firms with well-functioned board structure and better financial performance.

However, differing from diverse institutional investors, individual investors seem to prefer trading or holding small-scale electronic stocks. We deduce that individual investors might not have abundant funds to buy heavy-weighted stocks and high-priced stocks. As a result, they might trade and hold small-scale electronic stocks with the characteristics of high price volatility and speculation, which might result in not easily profiting on day trading stocks for individual investors.

## Discussion

In general, panel data models should be used more appropriate for firm-year observations than traditional linear models. Panel data normally submits data containing time series observations of a number of individuals [88] and offers researchers a large number of data points with lowering collinearity among explanatory variables, which enhance the efficiency of econometric estimates [89]. However, as disclosed that the biases in the standard errors vary widely and even are incorrect in many cases [84], we employ Petersen models in this study for grasping the relative accuracy, which would be beneficial for the robustness of the empirical results.

Nevertheless, for the robustness of the empirical results, we also employ the traditional panel data models (because of employing firm-year observations) and censored panel data models (due to concerning the characteristics of dependent variables) in this study. The results derived by adopting aforementioned models are similar to those by employing Petersen Models as shown in Tables 3 and 4. These findings indicate that our revealed results are robust after concerning different models employed. We then explain our revealed results with robustness concerns instead of presenting similar results by incorporating four more tables in the context.

## Conclusion

At the beginning of 2016, the day trading of stocks is expanded considerably in Taiwan, which arouses our interest to investigate whether the shareholdings and trading volumes of diverse investors would be affected by day trading ratio and turnover ratio. Therefore, we organize this research to explore these issues by incorporating several variables, such as board structure, financial statements, and other as controlling variable. This study includes FIIs, DIIs, SDs, and individual investors as diverse investors and reveals several impressive findings in return.

First, we find that FIIs would not prefer holding nor trading stocks with higher day trading ratios and turnover ratios, whereas individual investors prefer holding the stocks with higher day trading ratios. We infer that the results might stem from the fundamentals and the speculative concerns of these investors. In fact, to some extent, institutional investors may suffer the pressure of making profits and avoiding losses [90,91]. Meanwhile, institutional investors endure the pressure of pursuing higher performance [92] that arises from the extraordinary competition among institutional investors and mutual funds [93,94]. As a result, institutional investors should pay more attention to the fundamentals [95], as compared with the speculative concern of some individual investors [96].

Second, we show that day trading stocks through spot trading might crowd out day trading stocks through credit trading after the relaxation of day trading regulation. In fact, due to not

having abundant capital [97], individual investors in Taiwan day trade stocks through credit trading before the relaxation [98]. However, after the relaxation, day trading stocks through margin trading might be substituted by those through spot trading for individual investors. We speculate that this substitution might result from either spot day trading permitted after this relaxation or lower transaction cost of spot day trading. Thus, we argue that the substitution effect between margin day trading and spot day trading might explain this finding.

Third, institutional investors, including DIIs and SDs, would prefer trading the stocks with higher turnover ratios but FIIs are still lack of interest in trading these stocks. In addition, we argue that DIIs and SDs might not hold the stocks for the long run since they prefer holding shares with higher turnover ratios [99]. However, due to the evidence that FIIs are regarded as the successful investors in Taiwan [100] and even in East Asia [63,101], we argue that our findings may help investors realize more about the behaviors of diverse investors, especially the FIIs, after day trading relaxation in Taiwan.

Four, we reveal that the shareholding and trading behaviors of institutional investors are different from those of individual investors. This discovery is somewhat consistent with the previous studies [93,102], which might result in that the performance of institutional investors is better than those of individual investors [30,31]. In sum, we argue that our revealed results might be beneficial for individual investors. If individual investors take our findings into account when investing, they might generate proper decisions on whether aggressively trade or hold the stocks with higher day trading and turnover ratios [86,103,104].

By revealing that the shareholding and trading behaviors would be different among diverse investors, we claim that these revealed results might provide several implications that enrich the existing literature. First, we illustrate that investors might observe and even trace the behaviors of successful investors, such as FIIs, to enhance their investment performance, especially after the relaxation of day trading regulation in Taiwan. Furthermore, aside from focusing on the effects of day trading or turnover ratio on the shareholding and trading behaviors of diverse investors, we might have more proxies, such as accelerating turnover ratio except day trading ratio proposed in this study, to conduct more researches. Second, in addition to the relaxation of day trading regulation, we argue that corporate governance, financial performance, and firm scale should be concerned because FIIs prefer holding or trading the stocks probably based on these issues.

As with all research, this study has some limitations, which provides the direction for future research. First, by utilizing the data of TWSE listed firms due to the limited availability of data resources, the revealed results of this study might be likely twisted. Since the scales of TWSE listed firms are somewhat small, the values of these firms might be affected by the short-term large capital inflows and outflows. Second, as an emerging market of TWSE, the empirical results of this study might be different from those of developed markets. For future research, we should have more representative markets and find other crucial variables to conduct further studies related to affect the shareholding and trading of diverse investors.

## Supporting information

**S1 Data.**
(RAR)

## Author Contributions

**Conceptualization:** Wan-Hsiu Cheng, Yensen Ni.

**Data curation:** Ting-Hsun Ho, Chia-Jung Chiang.

**Methodology:** Wan-Hsiu Cheng, Yensen Ni, Paoyu Huang.

**Software:** Wan-Hsiu Cheng, Yensen Ni, Ting-Hsun Ho, Chia-Jung Chiang.

**Validation:** Paoyu Huang.

**Writing – original draft:** Wan-Hsiu Cheng, Yensen Ni, Ting-Hsun Ho, Chia-Jung Chiang, Paoyu Huang, Yirung Cheng.

**Writing – review & editing:** Wan-Hsiu Cheng, Yensen Ni, Ting-Hsun Ho, Chia-Jung Chiang, Paoyu Huang, Yirung Cheng.

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
