## [Decision Letter · Decision Letter 0]

6 Jan 2021

PONE-D-20-35453

Are the shareholding and trading behaviors of diverse investors affected by the relaxation of day trading?

PLOS ONE

Dear Dr. Ni,

Thank you for submitting your manuscript to PLOS ONE. After careful consideration, I feel that it has merit but does not fully meet PLOS ONE’s publication criteria as it currently stands. Therefore, I invite you to submit a revised version of the manuscript that addresses the points raised during the review process.

I agree with reviewers that in general this is an interesting paper to be considered for publication in Plos One. However, the I find that there is a lack of motivation in the research and the theoretical contribution of the manuscript is not justified.

On the other hand, reviewers also show serious concerns about the model used: variables, robustness, limitations, etc.

I suggest also to the authors to revise other minor questions commented by the reviewers, such us the references, introduction or the use of too long sentences through the manuscript.

According with our publication criteria this manuscript is not suitable of publication in its current version.

We look forward to receiving your revised manuscript.

Kind regards,

J E. Trinidad Segovia

Academic Editor

PLOS ONE

Journal Requirements:

Reviewers' comments:

Reviewer's Responses to Questions

**Comments to the Author**

1. Is the manuscript technically sound, and do the data support the conclusions?

Reviewer #1: Yes

Reviewer #2: Partly

Reviewer #3: Yes

2. Has the statistical analysis been performed appropriately and rigorously? 

Reviewer #1: Yes

Reviewer #2: Yes

Reviewer #3: No

3. Have the authors made all data underlying the findings in their manuscript fully available?

Reviewer #1: No

Reviewer #2: No

Reviewer #3: Yes

4. Is the manuscript presented in an intelligible fashion and written in standard English?

Reviewer #1: Yes

Reviewer #2: Yes

Reviewer #3: No

5. Review Comments to the Author

Reviewer #1: I have read this paper carefully. I believe there are some rather serious issues that must be addressed adequately.

1. The theoretical contribution you mentioned in the Introduction is not theoretically justified. There is not much information about what has been researched and what has not. The Introduction is largely occupied with background information which is not really helpful for developing your theoretical arguments. Thus, it is unclear where you position your research in the literature.

2. I hardly see how the three strands of literature are integrated to underpin your conceptual model. There is no clear articulation between the theories and your model and variables.

3. The conceptual development is still weak. You paper appears to be more like data-driven rather than fundamentally theory-driven. There is no clear Theories and Hypothesis Development" in the section 2.

4. Literature needs to be updated. It is not clear which literature body your paper may contribute to.

5. There are some issues in the empirics too. Authors think that the endogeneity problem might not be serious in this research. However, the statistic tests for the IV 2SLS and GMM are not presented. And, I do not think using IV 2SLS and GMM as well as Hausman statistics can prove the two independent variables are exogenous.

6. There should be a separate section for discussions with a focus on how your findings may make theoretical contributions.

7. If this paper aims to assess the policy effect of 2016 policy, a PSM-DID model may be a better model to deal with the obvious selection biases.

Reviewer #2: Title: Are the shareholding and trading behaviors of diverse investors affected by the relaxation of day trading?

Summary

The authors explore whether day trading volume over total trading volume and turnover ratio likely enhanced by an increase in day trading volume would affect the shareholding and trading behaviors of diverse institutional investors and even individual investors? The results indicate that; 1) foreign institutional investors would not prefer either holding or trading stocks with high day trading ratios, whereas individual investors would prefer holding such stocks. 2) domestic institutional investors and security dealers, instead of foreign institutional investors, would prefer trading stocks with high turnover ratios, but foreign institutional investors still lack interest in trading such stocks, implying that investing strategies would be diverse among various institutional investors. The study also provides some implications based on findings.

Overall, the paper considers an exciting and vital area and writeup is satisfactory. However, some improvements are suggested below.

1. It is always preferable to provide a paper for review with line numbering. In the absence of line numbering, it is hard to suggest correction exactly where needed.

2. The whole manuscript contains many wordy sentences. Such sentences may be split into small untestable pieces.

3. Is it mandated by the PLOS ONE author guidelines to abbreviate Literature Review section heading as “Literature Rev.”? if not it may be written fully.

4. The intext citations suffer some problems. Several citations are not consistent with the guidelines of PLOS ONE. Citations at the end of sentences are correctly done, while direct citations are not. For example, in literature review section Chan, Chockalingam, and Lai, McInish and Wood, Patell and Wolfson, Stephan and Whaley, and many more throughout the manuscript are not aligned with the citation guidelines. Authors may take care of these issues.

5. Defining abbreviations may be fixed carefully. For example, TWSE is not defined and directly used. However, later, defined in Descriptive statistics section. Similarly, Taiwan Economic Journal (TEJ) is defined twice? Once here and second in Descriptive statistics section.

6. Data stationarity condition is one of the basic for regression, wither time series estimator is used or panel. May the authors take into consideration the first- and second-generation panel unit-root and report the results as core Table inside the text and use those results as a predecessor to select the model to be estimated.

7. Detection and dealing the outliers is initial stage activity, which may not be reported and discuss after estimating VIF. It is surprising to me how 15 regressors are free from multicollinearity. Thus, it is suggested to report VIF, and also Hansen test results as appendices, to help understand readership about the data properties.

8. P12, authors stated that “……owing to the defects of the panel data models proposed by Petersen [72], we, therefore, employ the model proposed by Petersen for grasping the relative accuracy after taking the structure of the data into account.”. May the mentioned defect be highlighted, and how these can mislead the estimation? Moreover, it is expected to provide detailed advantages of the Petersen model opted by the authors. What other related methods are available in the field and how the selected model is superior over all of them?

9. Petersen (2008) have outlined comprehensively on how to cluster the standard error in various cases. Therefore, the authors are expected to articulate the selection of model taking guide from Petersen (2008).

10. I would be interested to know how Petersen (2008) deals with cross-sectional dependence in used panel dataset. By the way, have authors tested cross-sectional dependence? If yes, what were the results?

11. Inconsistent decimal places in Table 2 and subsequent Tables may be streamlined. It is standard in our field to use up to three decimal places. The word table may be written as “Table” intext where it is referred.

12. I guess line spacing is not aligned with rest of the document, and Petersen, 2009 departs from PLOS ONE reference style.

13. May the authors explain the reason behind relatively high constant in 1A, 3A? why it is equally low in the other two models?

14. Discussion section preferably be separated as independent section before the conclusion section.

15. The research touches an interesting area, while results may be supported by performing some robustness (an alternative measure of dependent/independent variable, alternative estimator, out of sample analysis or reduced sample analysis, or maybe with additional control). It is suggested to create a section before discussion as Robustness. The authors do have the liberty to perform at least any two robustness out of few suggested.

16. The sub-heading “Research Implications and further studies” may be omitted”. It is sufficient to discuss implications and future directions as plain text.

17. The first sentence of conclusion consists of almost 5 lines, which would be confusing to the readers. May authors carefully split such wordy sentence into small untestable sentences throughout the manuscript.

18. Few lines may be added on limitations of the current piece of research; then future directions will be adequate.

Reviewer #3: Using data on companies listed on Taiwan Stock Exchange (TWSE) over the 2016-2017 period, this paper examines the effect of the day trading ratio and turnover ratio on shareholding and trading behaviours of individual as well as domestic and international institutional investors. The topic is interesting, and the authors have produced some interesting results. The paper needs some improvements in my opinion as follows:

1. The motivation needs to be strengthened.

2. The main estimated coefficients need to be interpreted.

3. It may be useful to test for the presence of significant heteroscedasticity.

4. To establish robustness, it may be useful to also apply an alternative estimation technique.

5. The wiring needs improvement.

6. PLOS authors have the option to publish the peer review history of their article (what does this mean?). If published, this will include your full peer review and any attached files.

Reviewer #1: No

Reviewer #2: No

Reviewer #3: No

---

## [Author Response · Author response to Decision Letter 0]

23 Feb 2021

Reviewer #1

1. The theoretical contribution you mentioned in the Introduction is not theoretically justified. There is not much information about what has been researched and what has not. The Introduction is largely occupied with background information which is not really helpful for developing your theoretical arguments. Thus, it is unclear where you position your research in the literature.

Response to valuable comment:

We really appreciate your treasured comment. We agree with you that we should express clearly what has been researched in this study. Therefore, we generate some supplements to enhance our research theoretically in the literature. The modification is shown in the first three paragraphs of the Introduction from P. 2 to P. 3.

The following contents are shown in the context of this paper (P. 2-P. 3):

From the perspective of psychology, there is an environment-behavior theory that discusses the relationship between environmental change and behavior reaction. In theory, the environment is defined as the self-report measures used to assess subjective reactions to that environment and then people evaluate their behavior with regard to the environment accordingly. Besides, the intention is the best predictor of environmental behavior, whereas personal norms, attitudes, and perceived behavioral control have a significant influence on behavioral intention. In sum, the reaction is an attempt to find the results that make changes to the individual as a new thing and to increase knowledge. However, it seems to exist an interesting line of research links psychological influences and financial markets, which evokes our curiosity whether this psychological theory is still functioning for the investment activity. If it does work, we then might establish some investment models for investors to take suitable strategies in advance.

Inspired by the evidence that game rule change might cause behavioral change, we consider that it would be worthwhile to investigate the issue of environment-behavior change in the finance area. We speculate that the regulation change probably results in the change of investor’s behavior in return. On Feb. 1, 2016, Taiwanese government permits to unwind the regulation of day trading in stock market, which makes market participants to believe that the liquidity of market may be improved and the trade of stocks may have more flexibility. Even further, the relaxation may reduce the risk of overnight shareholdings for investors and may protect the interests of investors. However, we argue that there are some issues still unanswered after the relaxation. For example, whether individual and institutional investors, including foreign institutional investors, domestic institutional investors, and security dealers, would prefer trading or holding the stocks with higher day trading ratios, even higher turnover ratios? Does any difference exist for the preference of trading or holding these stocks among individual and institutional investors? These unresolved issues arouse our interest to organize the research.

We conduct this study to examine whether the relaxation of day trading regulation would change the diverse investors' behaviors, which is stemmed from the following motivations. First, we attempt to identify if the speculative individual investors might prefer investing stocks with higher day trading ratios and, in contrast, if the other individual deemed as the uninformative traders might not prefer trading these stocks due to the difficulty of making profits. Second, as realized with making more effort on fundamentals instead of speculation, institutional investors should tend to decrease the shareholdings and trading volume of stocks with higher day trading ratios. However, some institutional investors would prefer trading and holding the stocks with higher day trading ratios, especially for those having unreleased information. Therefore, we endeavor to solve the puzzle of whether these institutional investors have this preference due to liquidity concerns.

2. I hardly see how the three strands of literature are integrated to underpin your conceptual model. There is no clear articulation between the theories and your model and variables.

Response to valuable comment: 

Thank you for your valuable comment. We agree that we should have more description on the articulation among the three strands of literature and the conceptual model. In this study, we are interested in the reactions of investors’ behaviors after the relaxation of day trading regulation in Taiwan, which is inspired by the environment-behavior theory in psychology. Thus, we conduct the literature review for this study with the diverse investors and day trading as the issues to achieve this goal. Consequently, we add the inspiration and the aim of this study in the first two paragraphs and the third paragraph of the Introduction, respectively. Hopefully, these supplements can link the relations more tightly with the three strands of literature and the conceptual model. The modified text is shown on P. 2 and P. 3.

The following contents are shown in the context of this paper (P. 2 & P. 3):

From the perspective of psychology, there is an environment-behavior theory that discusses the relationship between environmental change and behavior reaction. In theory, the environment is defined as the self-report measures used to assess subjective reactions to that environment and then people evaluate their behavior with regard to the environment accordingly. Besides, the intention is the best predictor of environmental behavior, whereas personal norms, attitudes, and perceived behavioral control have a significant influence on behavioral intention. In sum, the reaction is an attempt to find the results that make changes to the individual as a new thing and to increase knowledge. However, it seems to exist an interesting line of research links psychological influences and financial markets, which evokes our curiosity whether this psychological theory is still functioning for the investment activity. If it does work, we then might establish some investment models for investors to take suitable strategies in advance.

Inspired by the evidence that game rule change might cause behavioral change, we consider that it would be worthwhile to investigate the issue of environment-behavior change in the finance area. We speculate that the regulation change probably results in the change of investor’s behavior in return. On Feb. 1, 2016, Taiwanese government permits to unwind the regulation of day trading in stock market, which makes market participants to believe that the liquidity of market may be improved and the trade of stocks may have more flexibility. Even further, the relaxation may reduce the risk of overnight shareholdings for investors and may protect the interests of investors. However, we argue that there are some issues still unanswered after the relaxation. For example, whether individual and institutional investors, including foreign institutional investors, domestic institutional investors, and security dealers, would prefer trading or holding the stocks with higher day trading ratios, even higher turnover ratios? Does any difference exist for the preference of trading or holding these stocks among individual and institutional investors? These unresolved issues arouse our interest to organize the research.

We conduct this study to examine whether the relaxation of day trading regulation would change the diverse investors’ behaviors, which is stemmed from the following motivations. First, we attempt to identify if the speculative individual investors might prefer investing stocks with higher day trading ratios and, in contrast, if the other individual deemed as the uninformative traders might not prefer trading these stocks due to the difficulty of making profits. Second, as realized with making more effort on fundamentals instead of speculation, institutional investors should tend to decrease the shareholdings and trading volume of stocks with higher day trading ratios. However, some institutional investors would prefer trading and holding the stocks with higher day trading ratios, especially for those having unreleased information. Therefore, we endeavor to solve the puzzle of whether these institutional investors have this preference due to liquidity concerns.

3. The conceptual development is still weak. Your paper appears to be more like data-driven rather than fundamentally theory-driven. There is no clear Theories and Hypothesis Development" in section 2.

Response to valuable comment: 

We truly appreciate your constructive comments. We agree that we should develop the theory and hypothesis in this study. Therefore, we supplement the theory in the first two paragraphs of the Introduction and the hypothesis in the Literature review section. The revision is presented on P. 2 and P. 6 to P. 9.

The following contents are shown in the context of this paper (P. 2 & P. 6-P. 9):

P. 2

From the perspective of psychology, there is an environment-behavior theory that discusses the relationship between environmental change and behavior reaction. In theory, the environment is defined as the self-report measures used to assess subjective reactions to that environment and then people evaluate their behavior with regard to the environment accordingly. Besides, the intention is the best predictor of environmental behavior, whereas personal norms, attitudes, and perceived behavioral control have a significant influence on behavioral intention. In sum, the reaction is an attempt to find the results that make changes to the individual as a new thing and to increase knowledge. However, it seems to exist an interesting line of research links psychological influences and financial markets, which evokes our curiosity whether this psychological theory is still functioning for the investment activity. If it does work, we then might establish some investment models for investors to take suitable strategies in advance.

Inspired by the evidence that game rule change might cause behavioral change, we consider that it would be worthwhile to investigate the issue of environment-behavior change in the finance area. We speculate that the regulation change probably results in the change of investor’s behavior in return. On Feb. 1, 2016, Taiwanese government permits to unwind the regulation of day trading in stock market, which makes market participants to believe that the liquidity of market may be improved and the trade of stocks may have more flexibility. Even further, the relaxation may reduce the risk of overnight shareholdings for investors and may protect the interests of investors. However, we argue that there are some issues still unanswered after the relaxation. For example, whether individual and institutional investors, including foreign institutional investors, domestic institutional investors, and security dealers, would prefer trading or holding the stocks with higher day trading ratios, even higher turnover ratios? Does any difference exist for the preference of trading or holding these stocks among individual and institutional investors? These unresolved issues arouse our interest to organize the research.

P. 6-P. 9

H1. The shareholding and trading behaviors of investors would be affected by the enhanced day trading volume after the relaxation of day trading regulation.

H2. The shareholding and trading behaviors of individual and institutional investors would be impacted by the relaxation of day trading regulation.

H3. The shareholding and trading behaviors of institutional investors would be affected by the relaxation of day trading regulation.

H4. The margin level and credit trading would be affected by the relaxation of day trading regulation.

H5. The behaviors of individual investors would be related to the issues of corporate governance and financial statement.

4. Literature needs to be updated. It is not clear which literature body your paper may contribute to.

Response to valuable comment: 

Thank you for your precious comment. We agree that we should review the articles cited in the literature to connect with the content of this study more closely. Hence, we adopt several updated papers and drop out unrelated citations. The revised paragraphs are revealed on each subsection of the Literature review and the updated references are displayed from P. 21 to P. 25.

The following contents are shown in the context of this paper:

P. 4- P. 5

Intraday trading and day trading

Intraday traders buy and sell financial instruments in the short term, typically within the same trading day [40]. In other words, intraday trading has an open position that is not left overnight [41] and its observations seem suitable for examining the volume–volatility relation [42]. The intraday data allows us to measure the immediate impact of a certain issue on stock returns and provides a finer picture that cannot be readily extracted from aggregating daily observations. In fact, the advantage of intraday information is particularly valuable if a certain issue influences stock returns more significantly at certain trading hours, for example, at the market opening periods [5].

Day trading, defined as the purchase and sale of the same stock by the same investor on the same day [15], has garnered growing attention in the investment world, especially from investors who are attracted to the prospect of earning high profits by investing only a few hours per day [43]. Although day trading has gained extensive popularity among investors, its profit potential remains a controversial issue [44]. Greater day trading activity leads to greater return volatility and the impact of a day trading shock dissipates gradually within an hour [45]. Besides, day trading increases the bid-ask spread, price depth, and stock volatility, indicating that day trading activities not only cause higher transaction costs and trading risk but also raise the market’s ability to absorb price impact. In sum, the impacts of stock day trading on market quality are not all positive. [46].

P. 6

Institutional and individual investors 

Due to having perverse security selection abilities and limited information, individual investors often sell the winning investments, hold the losing investments, make undiversified stock portfolios, and engage in repeating the past behaviors [52]. In fact, most of the trading losses for individual investors could be traced to their aggressive orders which likely cause the individual investors to suffer losses in stock markets [30]. On average, individual investors lose money from trading [15]. In general, they buy stocks that earn subpar returns and sell stocks that earn strong returns [53]. However, some individual investors might adjust their behavior, which improves their investment performance consequently [54]. 

Essentially, institutional investors consisted of foreign institutional investors (FIIs), domestic institutional investors (DIIs), and security dealers (SDs), act as vital roles in the business world [55].

P. 7-P.8

Diverse investors and day trading

Although individual day traders provide market liquidity by reducing the bid‐ask spread, temporary price volatility, and the temporal price impacts, most of them tend to behave as irrational contrarian traders [67]. As a result, individual investor trading results in systematic and economically large losses [30]. Aside from their aggressive orders as mentioned in the previous subsection, a large proportion of individual investors are underdiversified, and the extent of under-diversification is greater for investors who hold only retirement accounts [68]. Particularly, domestic individuals, accounting for the largest portion of total day trading activity, face substantial losses from day trading and individual day traders who trade more frequently and heavily are more likely to suffer such losses. On the contrary, domestic money managers and foreign institutional investors generally make substantial profits through day trading [69].

P. 21-P. 25

1. Clitheroe Jr, H. C.; Stokols, D.; Zmuidzinas, M. Conceptualizing the context of environment and behavior. J. Environmental Psychology. 1998, 18(1), 103-112.

2. Moradhaseli, S.; Ataei, P.; Norouzi, A. Analysis of students’ environmental behavior in the Agriculture College of Tarbiat Modares University, Iran (application of Planned Behavior Theory). J. Human Behavior in the Social Environment. 2017, 27(7), 733-742.

3. Gkargkavouzi, A.; Halkos, G.; Matsiori, S. Environmental behavior in a private-sphere context: Integrating theories of planned behavior and value belief norm, self-identity and habit. Resources, Conservation and Recycling. 2019, 148, 145-156.

4. Basri, H.;Amin, S.; Umiyati, M.; Mukhlis, H.; Irviani, R. Learning theory of conditioning. Learning. 2020, 7(8), 2024-2031.

5. Chang, S. C.; Chen, S. S.; Chou, R. K.; Lin, Y. H. Weather and intraday patterns in stock returns and trading activity. J. Banking Financ. 2008, 32(9), 1754-1799.

40.Baralis, E.; Cagliero, L.; Cerquitelli, T.; Garza, P.; Pulvirenti, F. Discovering profitable stocks for intraday trading. Information Sciences. 2017, 405, 91-106.

41.Xin, L.; Lam, K.; Philip, L. H. Effectiveness of filter trading as an intraday trading rule. Stud. Econ. Financ. 2019.

42.Darrat, A. F.; Rahman, S.; Zhong, M. Intraday trading volume and return volatility of the DJIA stocks: A note. J. Banking Financ. 2003, 27(10), 2035-2043.

43.Liew, P. X.; Lim, K. P.; Goh, K. L. Does proprietary day trading provide liquidity at a cost to investors?. Int. Rev. Financ. Anal. 2020, 68, 101455.

44.Kuo, S. H.; Tu, T. T. The Profitability of Day Trading and the Characteristics of Traders: Evidence from the Taiwan Futures Market. Int. Rev. Accounting, Banking Financ. 2020, 12(3), 16.

45.Chung, J. M.; Choe, H.; Kho, B. C. The impact of day‐trading on volatility and liquidity. Asia‐Pacific J. Financ. Stud. 2009, 38(2), 237-275.

46.Yang, T. Y.; Huang, S. Y.; Tsai, W. C.; Weng, P. S. The impacts of day trading activity on market quality: evidence from the policy change on the Taiwan stock market. J. Derivatives Quant. Stud. 2020, 28(4), 191-207.

53.Odean, T. Do investors trade too much?. American Econ. Rev. 1999, 89(5), 1279-1298.

55.Huang, P.; Ni, Y.; Cheng, Y. Does “Resetting” by Changing Corporate Name or Industry Category Appeals to Institutional Investments?. Contemporary Manage. Res. 2019, 15(3), 175-203.

61. Ferreira, M. A.; Massa, M.; Matos, P. Shareholders at the gate? Institutional investors and cross-border mergers and acquisitions. Rev. Financ. Stud. 2010, 23(2), 601-644.

68. Goetzmann, W. N.; Kumar, A. Equity portfolio diversification. Rev. Financ. 2008, 12(3), 433-463.

87. Hsiao, C. Panel data analysis—advantages and challenges. Test. 2007, 16(1), 1-22.

88. Hsiao, C. Analysis of panel data (No. 54). Cambridge University Press, Cambridge. 2014.

5. There are some issues in the empirics too. Authors think that the endogeneity problem might not be serious in this research. However, the statistic tests for the IV 2SLS and GMM are not presented. And, I do not think using IV 2SLS and GMM as well as Hausman statistics can prove the two independent variables are exogenous.

Response to valuable comment: 

We truly appreciate your constructive comments. Because of turnover ratio affected by firm scales, we then use firm scales as an instrument variable, which might imply that turnover ratio would be exogenous as shown insignificant Huusman tests by employing 2SLS and GMM approaches. Similar results are shown for daytrading ratio as well. In order to save space, we then explain here. 

ivregress gmm foreingtrading daytradingratio (turnoverratio = firmscales) director bigten manager pledge ceoduality boardsize indepdummy debtratio assetturnoverratio netprofitratio cashflowratio electronicdm, vce(unadjusted) 

est store IV_reg

reg foreingtrading daytradingratio turnoverratio director bigten manager pledge ceoduality boardsize indepdummy debtratio assetturnoverratio netprofitratio cashflowratio electronicdm

est store LS_reg

hausman IV_reg LS_reg

 ---- Coefficients ----

 | (b) (B) (b-B) sqrt(diag(V_b-V_B))

 | IV_reg LS_reg Difference S.E.

-------------+----------------------------------------------------------------

turnoverra~o | 243.1099 -.9550023 244.0649 105.7559

daytrading~o | -157.6539 -11.54232 -146.1116 142.2766

 director | 2.743037 -.1354781 2.878515 1.289202

 bigten | 3.147636 -.0101374 3.157773 1.680155

 manager | 3.230182 .0077809 3.222401 1.428217

 pledge | -.0246466 -.0547114 .0300648 .2601561

 ceoduality | -29.37603 -2.292998 -27.08303 16.46556

 boardsize | 11.77506 1.52628 10.24878 4.654944

indepdummy | 29.94947 -1.56913 31.5186 23.65889

 debtratio | .1479116 .0871041 .0608076 .2656556

assetturno~o | -22.89333 3.555542 -26.44887 13.86496

 netprofitr~o | -.09701 .0028976 -.0999076 .1328993

cashflowra~o | -.0131763 .0303554 -.0435316 .1241028

 electronicdm | -28.79385 4.457176 -33.25103 21.47277

 b = consistent under Ho and Ha; obtained from ivregress

 B = inconsistent under Ha, efficient under Ho; obtained from regress

 Test: Ho: difference in coefficients not systematic

 chi2(15) = (b-B)'[(V_b-V_B)^(-1)](b-B)

 = 6.43

 Prob>chi2 = 0.9717

ivregress 2sls foreingtrading daytradingratio (turnoverratio = firmscales) director bigten manager pledge ceoduality boardsize indepdummy debtratio assetturnoverratio netprofitratio cashflowratio electronicdm, first 

est store IV_reg

reg foreingtrading daytradingratio turnoverratio director bigten manager pledge ceoduality boardsize indepdummy debtratio assetturnoverratio netprofitratio cashflowratio electronicdm

est store LS_reg

hausman IV_reg LS_reg

 ---- Coefficients ----

 | (b) (B) (b-B) sqrt(diag(V_b-V_B))

 | IV_reg LS_reg Difference S.E.

-------------+----------------------------------------------------------------

 turnoverra~o | 243.1099 -.9550023 244.0649 122.2036

 daytrading~o | -157.6539 -11.54232 -146.1116 85.25695

 director | 2.743037 -.1354781 2.878515 1.476725

 bigten | 3.147636 -.0101374 3.57773 1.934468

 manager | 3.230182 .0077809 3.222401 1.661387

 pledge | -.0246466 -.0547114 .0300648 .2966378

 ceoduality | -29.37603 -2.292998 -27.08303 17.03619

 boardsize | 11.77506 1.52628 10.24878 5.523451

 indepdummy | 29.94947 -1.56913 31.5186 22.73928

 debtratio | .1479116 .0871041 .0608076 .2792552

 assetturno~o | -22.89333 3.555542 -26.44887 15.75222

 netprofitr~o | -.09701 .0028976 -.0999076 .1085325

 cashflowra~o | -.0131763 .0303554 -.0435316 .0764813

 electronicdm | -28.79385 4.457176 -33.25103 20.22016

 b = consistent under Ho and Ha; obtained from ivregress

 B = inconsistent under Ha, efficient under Ho; obtained from regress

 Test: Ho: difference in coefficients not systematic

 chi2(15) = (b-B)'[(V_b-V_B)^(-1)](b-B)

 = 3.99

 Prob>chi2 = 0.9978

6. There should be a separate section for discussions with a focus on how your findings may make theoretical contributions.

Response to valuable comment: 

Thank you for your generous comments. In fact, we discuss the revealed results in the Empirical findings section directly. By doing so, we consider that we might express the denotations and the implications for the real world in time. Nevertheless, we do develop a Discuss section to discuss the models employed in this study for the purpose of robustness. The paragraphs are presented on P. 18 and P. 19.

The following contents are shown in the context of this paper (P. 18 & P. 19):

Discussion

In general, panel data models should be more appropriate to be used than traditional linear models for employing firm-year observations. Panel data normally submits data containing time series observations of a number of individuals and offers researchers a large number of data points with lowering collinearity among explanatory variables, which enhance the efficiency of econometric estimates. However, as disclosed that the biases in the standard errors vary widely and even are incorrect in many cases, we employ Petersen models in this study for grasping the relative accuracy, which would be beneficial for the robustness of the empirical results.

Nevertheless, for the robustness of the empirical results, we also employ the traditional panel data models (because of employing firm-year observations) and censored panel data models (due to concerning the characteristics of dependent variables) in this study. The results derived by adopting the aforementioned models are similar to those by employing Petersen Models as shown in Tables 3-4. These findings indicate that our revealed results are robust after concerning different models employed.

7. If this paper aims to assess the policy effect of 2016 policy, a PSM-DID model may be a better model to deal with the obvious selection biases.

Response to valuable comment:

We really appreciate your valuable comment. First, we understood the advantage of analyzing the policy effects using the PSM-DID model, however, we couldn’t compose the entire data due to the data constrained. Considering the stock market after the relaxing policy of day trading is being in a new stage and unreversed trend. We then tried to investigate the behavior relationships in a new stage after the policy relaxing. Secondly, we select the data for the years 2016 and 2017. As the news reported by TWSE, 907 stocks eligible for day trading account for 98.75% of the total market capitalization in TWSE after the relaxation of the day trading regulation. Thus, the selection biases are not mentioned here.

 

Reviewer #2

1. It is always preferable to provide a paper for review with line numbering. In the absence of line numbering, it is hard to suggest correction exactly where needed.

Response to valuable comment:

Thank you for your treasured comment. We conduct this study with double-spacing lines. By increasing the space between lines, we expect to have more easy readability for this study. 

2. The whole manuscript contains many wordy sentences. Such sentences may be split into small untestable pieces.

Response to valuable comment:

We truly appreciate your constructive comments. We agree that several sentences in this study are of long wordings. Therefore, we reedit the whole paper and make sentences easier and more comfortable to be read.

3. Is it mandated by the PLOS ONE author guidelines to abbreviate Literature Review section heading as “Literature Rev.”? if not it may be written fully.

Response to valuable comment:

Thank you for your constructive comment. As you suggested, we rewrite the section heading of the Literature review and the revision is displayed on P. 4.

The following contents are shown in the context of this paper (P. 4):

Literature Review and hypotheses proposed

4. The intext citations suffer some problems. Several citations are not consistent with the guidelines of PLOS ONE. Citations at the end of sentences are correctly done, while direct citations are not. For example, in literature review section Chan, Chockalingam, and Lai, McInish and Wood, Patell and Wolfson, Stephan and Whaley, and many more throughout the manuscript are not aligned with the citation guidelines. Authors may take care of these issues.

Response to valuable comment: 

Thank you for your valuable comment. As you commented, we have checked and revised the expression of citations for the whole study. 

5. Defining abbreviations may be fixed carefully. For example, TWSE is not defined and directly used. However, later, defined in Descriptive statistics section. Similarly, Taiwan Economic Journal (TEJ) is defined twice? Once here and second in Descriptive statistics section.

Response to valuable comment: 

We truly appreciate your constructive comments. We agree with you that we should define the terms used in this study in proper order. Thus, we state the definition of TWSE at the beginning of the Data subsection firstly, as shown on P. 10, and use the abbreviation on the Descriptive statistics section, as presented on P. 12. Simultaneously, we do the similar revision on the expression of TEJ, as displayed on P. 10.

The following contents are shown in the context of this paper (P. 10 & P. 12):

P. 10

Data

We collect daily data of day trading volume from the website of Taiwan Stock Exchange (TWSE) over the period of 2016-2017 due to the concern of data consistency.

(In the second paragraph)

…The former could be measured as mentioned above, and the latter could be derived from Taiwan Economic Journal (TEJ)…

P. 12

Descriptive statistics

By utilizing 1,577 observations of TWSE listed firms over the period of 2016-2017 as our samples, Table 2 presents the descriptive statistics, including the number of observations, means, medians, standard deviations, minima, and maxima of the variables employed in this study.

6. Data stationarity condition is one of the basic for regression, wither time series estimator is used or panel. May the authors take into consideration the first- and second-generation panel unit-root and report the results as core Table inside the text and use those results as a predecessor to select the model to be estimated.

Response to valuable comment: 

Thank you for your precious comment. According to the suggestion of the Stata manual, the manual explains that when the number of time periods, T, is small (less than 10 or 15), the test suffers from severe size distortions when fixed effects or time trends are included; in these cases, using altt (make sample adjustment to T) results in much improved size properties at the expense of significantly less power. As a result, we might not take into consideration the first- and second-generation panel unit-root due to employing two year data (i.e. T=2 far less than 10). In addition, differencing time series data such as stock prices and oil prices would be often necessary due to the concern of stationary; however, the movements of the board structure variables (e.g. director holding ratio) and investors holding variables (e.g. foreign holding ratio) in subsequent years might not the same as those of stock prices and oil prices due to stationary concern. 

7. Detection and dealing the outliers is initial stage activity, which may not be reported and discuss after estimating VIF. It is surprising to me how 15 regressors are free from multicollinearity. Thus, it is suggested to report VIF, and also Hansen test results as appendices, to help understand readership about the data properties.

Response to valuable comment: 

We truly appreciate your constructive comments. We then report the VIF after detecting and dealing with the outliers at the initial stage. The VIF values for these variables are less than three as shown below, which might indicate that these variables might be free from multicollinearity. 

Program for VIF tests: 

. reg foreingtrading daytradingratio turnoverratio director bigten manager pledge ceoduality boardsize indepdummy debtratio assetturnoverratio netprofitratio cashflowratio electronicdm firmscales, vce(robust)

. vif 

 Variable | VIF 1/VIF 

-------------+----------------------

daytrading~o | 2.72 0.367046

turnoverra~o | 2.18 0.459021

 electronicdm | 1.79 0.559053

cashflowra~o | 1.52 0.656257

 firmscales | 1.47 0.678484

 boardsize | 1.39 0.717155

 debtratio | 1.37 0.729033

 netprofitr~o | 1.36 0.734362

 manager | 1.30 0.769412

 director | 1.26 0.795768

 assetturno~o | 1.16 0.863615

 ceoduality | 1.07 0.931624

 indepdummy | 1.07 0.933312

 bigten | 1.06 0.941964

 pledge | 1.04 0.961842

-------------+----------------------

 Mean VIF | 1.45

8. P12, authors stated that “……owing to the defects of the panel data models proposed by Petersen [72], we, therefore, employ the model proposed by Petersen for grasping the relative accuracy after taking the structure of the data into account.”. May the mentioned defect be highlighted, and how these can mislead the estimation? Moreover, it is expected to provide detailed advantages of the Petersen model opted by the authors. What other related methods are available in the field and how the selected model is superior over all of them?

Response to valuable comment: 

Thank you for your generous comments. As you suggested, we address the drawback of panel data models in the Discussion section. To avoid the widely varying biases in the standard errors generated by panel data models, we employ Petersen models in this study and expect to enhance the robustness of the empirical results. The expression is presented on P. 18 and P. 19.

The following contents are shown in the context of this paper (P. 18 & P. 19):

Discussion

In general, panel data models should be more appropriate to be used than traditional linear models for employing firm-year observations. Panel data normally submits data containing time series observations of a number of individuals and offers researchers a large number of data points with lowering collinearity among explanatory variables, which enhance the efficiency of econometric estimates. However, as disclosed that the biases in the standard errors vary widely and even are incorrect in many cases, we employ Petersen models in this study for grasping the relative accuracy, which would be beneficial for the robustness of the empirical results.

9. Petersen (2008) have outlined comprehensively on how to cluster the standard error in various cases. Therefore, the authors are expected to articulate the selection of model taking guide from Petersen (2008).

Response to valuable comment:

We really appreciate your valuable comment. As mentioned in the previous comment, we employ Petersen models in this study to avoid the widely varying biases in the standard errors generated by panel data models. The description is revealed on P. 18 and P. 19.

The following contents are shown in the context of this paper (P. 18 & P. 19):

Discussion

In general, panel data models should be more appropriate to be used than traditional linear models for employing firm-year observations. Panel data normally submits data containing time series observations of a number of individuals and offers researchers a large number of data points with lowering collinearity among explanatory variables, which enhance the efficiency of econometric estimates. However, as disclosed that the biases in the standard errors vary widely and even are incorrect in many cases, we employ Petersen models in this study for grasping the relative accuracy, which would be beneficial for the robustness of the empirical results.

10. I would be interested to know how Petersen (2008) deals with cross-sectional dependence in used panel dataset. By the way, have authors tested cross-sectional dependence? If yes, what were the results?

Response to valuable comment:

Thank you for your treasured comment. In this study, instead of concerning cluster between year and firm (i.e. Petersen models), the model concerning the clustered between firms (i.e. concerning cross-sectional dependence) is suggested by Petersen (2008). We then reveal that the results are similar to those with concerning clustered between firm and year. We infer that similar findings might result from the results adjusted among clustered residuals only. In order to save space, we explain here. 

Program for the concern of cross-sectional dependence by concerning clustered between firms (i.e. adding robust cluster(firm) at the end of the following model as shown below. 

Stata Programming Instructions for Petersen website: https://www.kellogg.northwestern.edu/faculty/petersen/htm/papers/se/se_programming.htm

Clustered (Rogers) Standard Errors – One dimension

 regress dependent_variable independent_variables, robust cluster(cluster_variable)

regress foreingtrading daytradingratio turnoverratio director bigten manager pledge ceoduality boardsize indepdummy debtratio assetturnoverratio netprofitratio cashflowratio electronicdm firmscales, robust cluster(firm)

Clustered Standard Errors – Two dimensions

 cluster2 dependent_variable independent_variables, cluster(cluster_variable_one) tcluster(cluster_variable_two)

11. Inconsistent decimal places in Table 2 and subsequent Tables may be streamlined. It is standard in our field to use up to three decimal places. The word table may be written as “Table” intext where it is referred.

Response to valuable comment:

We truly appreciate your constructive comments. Concerning the consistent decimal places in Table 2, the subsequent Tables should be streamlined (i.e. up to three decimal places). However, due to 0.000 might be sometimes appeared if employing up to three decimal places only, we suggest that our empirical results might be properly presented up to five decimal places in Tables 3-4. Besides, the word table would be written as “Table” where it is referred to in this paper. Thank you for your suggestion for the consistency of the paper presentation. We are really appreciated it. 

12. I guess line spacing is not aligned with rest of the document, and Petersen, 2009 departs from PLOS ONE reference style.

Response to valuable comment:

Thank you for your constructive comment. As you mentioned, we have checked and revised the expression of citations for the whole study.

13. May the authors explain the reason behind relatively high constant in 1A, 3A? why it is equally low in the other two models?

Response to valuable comment:

We really appreciate your treasured comment. 

The value and range dependent variable as presented in Table 2 

14. Discussion section preferably be separated as independent section before the conclusion section.

Response to valuable comment: 

Thank you for your valuable comment. As you suggested, we develop a Discuss section to discuss the models employed in this study for the purpose of robustness. The paragraphs are presented on P. 18 and P. 19.

The following contents are shown in the context of this paper (P. 18 & P. 19):

Discussion

In general, panel data models should be more appropriate to be used than traditional linear models for employing firm-year observations. Panel data normally submits data containing time series observations of a number of individuals and offers researchers a large number of data points with lowering collinearity among explanatory variables, which enhance the efficiency of econometric estimates. However, as disclosed that the biases in the standard errors vary widely and even are incorrect in many cases, we employ Petersen models in this study for grasping the relative accuracy, which would be beneficial for the robustness of the empirical results.

Nevertheless, for the robustness of the empirical results, we also employ the traditional panel data models (because of employing firm-year observations) and censored panel data models (due to concerning the characteristics of dependent variables) in this study. The results derived by adopting the aforementioned models are similar to those by employing Petersen Models as shown in Tables 3-4. These findings indicate that our revealed results are robust after concerning different models employed.

15. The research touches an interesting area, while results may be supported by performing some robustness (an alternative measure of dependent/independent variable, alternative estimator, out of sample analysis or reduced sample analysis, or maybe with additional control). It is suggested to create a section before discussion as Robustness. The authors do have the liberty to perform at least any two robustness out of few suggested.

Response to valuable comment: 

We truly appreciate your constructive comments. we develop a Discuss section to discuss the models employed in this study for the purpose of robustness. The paragraphs are shown on P. 18 and P. 19.

The following contents are shown in the context of this paper (P. 18 & P. 19):

Discussion

In general, panel data models should be more appropriate to be used than traditional linear models for employing firm-year observations. Panel data normally submits data containing time series observations of a number of individuals and offers researchers a large number of data points with lowering collinearity among explanatory variables, which enhance the efficiency of econometric estimates. However, as disclosed that the biases in the standard errors vary widely and even are incorrect in many cases, we employ Petersen models in this study for grasping the relative accuracy, which would be beneficial for the robustness of the empirical results.

Nevertheless, for the robustness of the empirical results, we also employ the traditional panel data models (because of employing firm-year observations) and censored panel data models (due to concerning the characteristics of dependent variables) in this study. The results derived by adopting the aforementioned models are similar to those by employing Petersen Models as shown in Tables 3-4. These findings indicate that our revealed results are robust after concerning different models employed.

16. The sub-heading “Research Implications and further studies” may be omitted”. It is sufficient to discuss implications and future directions as plain text.

Response to valuable comment: 

Thank you for your precious comment. We adopt your suggestion and delete the sub-heading “Research Implications and further studies.” The revised paragraphs are displayed on P. 20 and P. 21.

The following contents are shown in the context of this paper (P. 20 & P. 21):

By revealing that the shareholding and trading behaviors would be different among diverse investors, we claim that these revealed results might provide several implications that enrich the existing literature. First, we illustrate that investors might observe and even trace the behaviors of successful investors, such as FIIs, to enhance their investment performance, especially after the relaxation of day trading regulation in Taiwan. Furthermore, aside from focusing on the effects of day trading or turnover ratio on the shareholding and trading behaviors of diverse investors, we might have more proxies, such as accelerating turnover ratio except day trading ratio proposed in this study, to conduct more researches. 

Second, in addition to the relaxation of day trading regulation, we argue that corporate governance, financial performance, and firm scale should be concerned because FIIs prefer holding or trading the stocks probably based on these issues.

As with all research, this study has some limitations, which provides the direction for future research. First, by utilizing the data of TWSE listed firms due to the limited availability of data resources, the revealed results of this study might be likely twisted. Since the scales of TWSE listed firms are somewhat small, the values of these firms might be affected by the short-term large capital inflows and outflows. Second, as an emerging market of TWSE, the empirical results of this study might be different from those of developed markets. For future research, we should have more representative markets and find other crucial variables to conduct further studies related to affect the shareholding and trading of diverse investors.

17. The first sentence of conclusion consists of almost 5 lines, which would be confusing to the readers. May authors carefully split such wordy sentence into small untestable sentences throughout the manuscript.

Response to valuable comment: 

We truly appreciate your constructive comments. We agree with you that the first sentence of the Conclusion is too wordy. Therefore, we reorganize this paragraph and separate the expression into three short sentences. The revised text is shown on P. 19.

The following contents are shown in the context of this paper (P. 19):

At the beginning of 2016, the day trading of stocks is expanded considerably in Taiwan, which arouses our interest to investigate whether the shareholding and trading volume of diverse investors would be affected by day trading ratio and turnover ratio. Therefore, we organize this research to explore these issues by incorporating several variables, such as board structure, financial statements, and other as controlling variable. This study includes FIIs, DIIs, SDs, and individual investors as diverse investors and reveals several impressive findings in return.

18. Few lines may be added on limitations of the current piece of research; then future directions will be adequate.

Response to valuable comment: 

Thank you for your generous comments. We adopt your suggestion and address the limitation and further research for this study. The paragraph is shown on P. 21.

The following contents are shown in the context of this paper (P. 21):

As with all research, this study has some limitations, which provides the direction for future research. First, by utilizing the data of TWSE listed firms due to the limited availability of data resources, the revealed results of this study might be likely twisted. Since the scales of TWSE listed firms are somewhat small, the values of these firms might be affected by the short-term large capital inflows and outflows. Second, as an emerging market of TWSE, the empirical results of this study might be different from those of developed markets. For future research, we should have more representative markets and find other crucial variables to conduct further studies related to affect the shareholding and trading of diverse investors.

 

Reviewer #3

1. The motivation needs to be strengthened.

Response to valuable comment:

We really appreciate your valuable comment. We agree with you that we should strengthen the motivation of this study because motivation is the original intention to conduct research. If the motivation is strong enough, researchers might generate high quality work that attracts people to cite. Hence, we reorganize the motivation paragraph in the Introduction section and attempt to improve the value of this study. The revision is displayed on P. 3.

The following contents are shown in the context of this paper (P. 3):

We conduct this study to examine whether the relaxation of day trading regulation would change the diverse investors' behaviors, which is stemmed from the following motivations. First, we attempt to identify if the speculative individual investors might prefer investing stocks with higher day trading ratios and, in contrast, if the other individual deemed as the uninformative traders might not prefer trading these stocks due to the difficulty of making profits. Second, as realized with making more effort on fundamentals instead of speculation, institutional investors should tend to decrease the shareholdings and trading volume of stocks with higher day trading ratios. However, some institutional investors would prefer trading and holding the stocks with higher day trading ratios, especially for those having unreleased information. Therefore, we endeavor to solve the puzzle of whether these institutional investors have this preference due to liquidity concerns.

2. The main estimated coefficients need to be interpreted.

Response to valuable comment:

Thank you for your treasured comment. We would endeavor to interpret the main estimated coefficients in this study. For example, Table 3 reveals that day trading ratio and turnover ratio negatively relate to the foreign shareholding ratio (��=-0.0189299**, P< 0.05; ��=-0.0035782, P<0.05) but positively relate to the market level ratio (��=0.0999011***, P<0.01; ��=-0.0388799, P<0.01). This result indicates that FIIs might not prefer holding the stocks with higher day trading ratio and turnover ratio but individual investors prefer holding these stocks.

3. It may be useful to test for the presence of significant heteroscedasticity.

Response to valuable comment:

We truly appreciate your constructive comments. 

After running the models without concerning heteroscedasticity issues, we then reveal that heteroscedasticity issues would be serious in our models as shown Breusch-Pagan / Cook-Weisberg test for heteroskedasticity as shown below

. estat hettest

Breusch-Pagan / Cook-Weisberg test for heteroskedasticity 

 Ho: Constant variance

 Variables: fitted values of foreingtrading

 chi2(1) = 412.37

 Prob > chi2 = 0.00001

As a result, due to significant heteroscedasticity existed in our models, we then use the comment vce(robust) recommend by Stata to deal with heteroscedasticity issues in our models. 

4. To establish robustness, it may be useful to also apply an alternative estimation technique.

Response to valuable comment:

Thank you for your constructive comment. In fact, instead of employing Petersen regression models, we also employ traditional panel data models and even censored panel data models for robustness. The results are similar to those revealed in this study. Because of concerning shareholding and trading behaviors of diverse investors, the length of this paper would be regarded as a long paper. We then explain the above concern in footnote 12. 

5. The wiring needs improvement.

Response to valuable comment:

Thank you for your generous comments. We have amended this study completely. We hope that this study is expressed clearer and easier to read after the modification.

---

## [Decision Letter · Decision Letter 1]

16 Mar 2021

PONE-D-20-35453R1

Are the shareholding and trading behaviors of diverse investors affected by the relaxation of day trading?

PLOS ONE

Dear Dr. Ni,

Thank you for submitting your manuscript to PLOS ONE. After careful consideration, I feel that it has merit but does not fully meet PLOS ONE’s publication criteria as it currently stands. Therefore, I invite you to submit a revised version of the manuscript that addresses the points raised during the review process.

Major concerns have been attended but one of the reviewers still requests answers about some minor questions. I suggest to the authors to attend or to explain why these comments are not addressed.

We look forward to receiving your revised manuscript.

Kind regards,

J E. Trinidad Segovia

Academic Editor

PLOS ONE

Journal Requirements:

Reviewers' comments:

Reviewer's Responses to Questions

**Comments to the Author**

1. If the authors have adequately addressed your comments raised in a previous round of review and you feel that this manuscript is now acceptable for publication, you may indicate that here to bypass the “Comments to the Author” section, enter your conflict of interest statement in the “Confidential to Editor” section, and submit your "Accept" recommendation.

Reviewer #1: All comments have been addressed

Reviewer #2: All comments have been addressed

Reviewer #3: All comments have been addressed

2. Is the manuscript technically sound, and do the data support the conclusions?

Reviewer #1: Yes

Reviewer #2: Yes

Reviewer #3: Yes

3. Has the statistical analysis been performed appropriately and rigorously? 

Reviewer #1: Yes

Reviewer #2: Yes

Reviewer #3: Yes

4. Have the authors made all data underlying the findings in their manuscript fully available?

Reviewer #1: Yes

Reviewer #2: No

Reviewer #3: Yes

5. Is the manuscript presented in an intelligible fashion and written in standard English?

Reviewer #1: Yes

Reviewer #2: No

Reviewer #3: Yes

6. Review Comments to the Author

Reviewer #1: Dear author(s)

Your revised manuscript addresses the concerns I raised in the first round of reviews. Good luck with your research.

Reviewer #2: 1. The point was to insert line numbering, while response is double-spacing which has nothing to do with line numbering. Line numbering helps referring exact point where authors may make correction. There are 5 authors in this paper and none of them understand the difference between line numbering and line spacing. Go to Layout menu, click on <line numbers=""> and chose <continuous>. I hope it will be understandable now to 5 authors.

11. There is no point to use diplomatic wording iteratively that “We truly appreciate your constructive comments” and do nothing to address the comment. In Table 2 if 0 may appear, so 0.000 may also appear which has same meanings. There are many numbers in 2 decimals, few in 3 others in none. Suppose we accept your argument of 5 decimal place, why same is not done in Table 2.?

13. Comment 13. It was asked to explain the reason behind relatively high constant in 1A, 3A? why it is equally low in the other two models? Which is responded in nonsense why that “The value and range dependent variable as presented in Table 2”. By the way what descriptive statistics has to do with justification of constant reported in other Tables?

The paper may be professionally copyedited.</continuous></line>

Reviewer #3: (No Response)

7. PLOS authors have the option to publish the peer review history of their article (what does this mean?). If published, this will include your full peer review and any attached files.

Reviewer #1: No

Reviewer #2: No

Reviewer #3: No

---

## [Author Response · Author response to Decision Letter 1]

27 Mar 2021

Reviewers' comments:

Reviewer #2

1. The point was to insert line numbering, while response is double-spacing which has nothing to do with line numbering. Line numbering helps referring exact point where authors may make correction. There are 5 authors in this paper and none of them understand the difference between line numbering and line spacing. Go to Layout menu, click on and chose. I hope it will be understandable now to 5 authors.

Thank you for your precious comment. We are so sorry for missing the line numbering on the previous revision. As you suggested, we insert the numbers of lines for the whole paper and hope to improve the readability. Again, we truly appreciate your warm reminding.

2. There is no point to use diplomatic wording iteratively that “We truly appreciate your constructive comments” and do nothing to address the comment. In Table 2 if 0 may appear, so 0.000 may also appear which has same meanings. There are many numbers in 2 decimals, few in 3 others in none. Suppose we accept your argument of 5 decimal place, why same is not done in Table 2.?

Response to valuable comment:

We deeply appreciate your treasured comment. We are so sorry for missing to modify the numbers in Table 2 on the previous revision. As you suggested, we revise the numbers in Table 2 in 5 decimals, except the observations. The modification is shown on P. 13 to P. 14. Once more, thanks for your kind reminding.

3. Comment 13. It was asked to explain the reason behind relatively high constant in 1A, 4A? why it is equally low in the other two models? Which is responded in nonsense why that “The value and range dependent variable as presented in Table 2”. By the way what descriptive statistics has to do with justification of constant reported in other Tables?

Response to valuable comment:

According to the regression model, the constant is often defined as the mean of the dependent variable when we set all of the independent variables in the model to zero. Thus, the dependent variable with high mean might have a relatively high constant. Besides, if the positive (negative) relationship exists between the dependent variable and most of the independent variables, the constant term is likely to be positive (negative) in the model. 

As a result, the means of foreign shareholding ratio and margin level ratio are higher than those of domestic institutional shareholding ratio and security dealer shareholding ratio, which is the reason to that the constants of 1A and 4A are higher than those of 2A and 3A. We then learn that descriptive statistics shown in Table 2 would be related to the constants reported in Tables 3. Besides, similar findings are shown in Table 4. Thank you so much for your valuable comment.

4. The paper may be professionally copyedited.

Response to valuable comment:

Thank you for your valuable comment. We have someone who is good in the English writing academically to reedit this paper. We hope the paper being easier to be read after the modification.

---

## [Editor Report · Decision Letter 2]

31 Mar 2021

Are the shareholding and trading behaviors of diverse investors affected by the relaxation of day trading?

PONE-D-20-35453R2

Dear Dr. Ni,

We’re pleased to inform you that your manuscript has been judged scientifically suitable for publication and will be formally accepted for publication once it meets all outstanding technical requirements.

Kind regards,

J E. Trinidad Segovia

Academic Editor

PLOS ONE
---

## [Editor Report · Acceptance letter]

5 Apr 2021

PONE-D-20-35453R2 

Are the shareholding and trading behaviors of diverse investors affected by the relaxation of day trading? 

Dear Dr. Ni:

I'm pleased to inform you that your manuscript has been deemed suitable for publication in PLOS ONE. Congratulations! Your manuscript is now with our production department. 

Kind regards, 

on behalf of

Dr. J E. Trinidad Segovia 

Academic Editor

PLOS ONE